# The Efficacy of Powered Oscillating Heads vs. Powered Sonic Action Heads Toothbrushes to Maintain Periodontal and Peri-Implant Health: A Narrative Review

**DOI:** 10.3390/ijerph18041468

**Published:** 2021-02-04

**Authors:** Camilla Preda, Andrea Butera, Silvia Pelle, Eleonora Pautasso, Alessandro Chiesa, Francesca Esposito, Giacomo Oldoini, Andrea Scribante, Anna Maria Genovesi, Saverio Cosola

**Affiliations:** 1Tuscan Stomatologic Institute, Versilia General Hospital, 55041 Lido di Camaiore, Italy; camilla.preda@unipv.it (C.P.); silviapelledh@gmail.com (S.P.); pautasso.e@gmail.com (E.P.); francesca.esposito@unipv.it (F.E.); giacomo.oldoini88@gmail.com (G.O.); anmgen@tiscali.it (A.M.G.); s.cosola@hotmail.it (S.C.); 2Study Center for Multidisciplinary Regenerative Research, Guglielmo Marconi University, 00100 Rome, Italy; alessandro.chiesa@unipv.it; 3Department of Clinical, Surgical, Diagnostic an Paediatric Sciences University of Pavia, 27100 Pavia, Italy; andrea.scribante@unipv.it; 4Department of Dentistry, Unicamillus International Medical University, 00100 Rome, Italy; 5Department of Stomatology, University of Valencia, 46001 Valencia, Spain

**Keywords:** rotating oscillating heads (ORHs), sonic action heads (SAHs), electric toothbrushes, home care, domiciliary oral hygiene

## Abstract

Objectives: To compare the efficacy of rotating-oscillating heads (ORHs) VS sonic action heads (SAHs) powered toothbrushes on plaque accumulation and gingival inflammation. Methods: An electronic (MEDLINE, Embase, Inspec, PQ SciTech and BIOSIS) and a complementary manual search were made to detect eligible studies. RCTs meeting the following criteria were included: final timepoint longer than 15 days; year of publication after 2000; patients without orthodontic appliances or severe systemic/psychiatric diseases. Studies comparing two or more different types of sonic/roto-oscillating toothbrushes were excluded. Selection of articles, extraction of data, and assessment of quality were made independently by several reviewers. Results: 12 trials (1433 participants) were included. The differences between ORHs and SAHs toothbrushes were expressed as weighted mean differences (WMD) and 95% confidence intervals (CI). The heterogeneity of data was evaluated. Concerning Plaque Index, both toothbrushes obtained comparable results. Six trials of up to 3 months and at an unclear risk of bias provided significant outcomes in terms of gingival inflammation in favor of ORHs toothbrush. Evidence resulting from three trials of up to 6 months and at a high/low risk of bias stated SAHs toothbrush superiority in gingival inflammation. Conclusions: Both ORHs and SAHs toothbrushes improved the outcomes measured from the baseline. In most of the good quality trials included, SAHs toothbrush showed statistical better long-term results. Due to the shortage of investigations, no further accurate conclusions can be outlined with reference to the superiority of a specific powered toothbrush over the other.

## 1. Introduction

It is well known that individual factors and the presence of bacterial plaque are associated with the development of gingivitis, periodontal disease, dental caries, and peri-implant inflammation.

The mechanical disruption of the bacterial plaque represents the main objective of oral hygiene and preventive dentistry [1].

Even if professional oral hygiene therapy is required to remove sub-gingival plaque and reduce gingival inflammation, recent studies have suggested that daily domiciliary patient-administered mechanical plaque control should be considered as the gold standard, and the most important factor for the management and reduction of plaque accumulation [2,3].

In this scenario, oral hygiene motivation, efficient therapeutic instruments, and the patient’s compliance play the main role in maintaining a healthy oral environment [4,5].

One of the major challenges for clinicians is to achieve patient motivation and to teach them a proper way to brush their teeth—but also not to make them forget these instructions as soon as they return home.

For these reasons, in recent years, different powered toothbrushes were introduced to the market with the aim to simplify patients’ daily oral care—avoiding long training periods and without requiring specific toothbrushing techniques [6,7,8].

Several systematic reviews investigated the efficacy of powered toothbrushes compared with manual toothbrushes, in terms of gingivitis control and clinical outcomes, such as bleeding on probing and the Plaque Index both around natural teeth and dental implants [9,10,11].

Researchers and clinicians agree on the advantages of daily oral hygiene managed with powered toothbrushes, as they seem to be more predictable than manual toothbrushes used with multiple techniques (in terms of both plaque and inflammation control, and in terms of collateral effects, such as gingival recession and dental abfraction.) [12,13].

The last two reviews on this topic (Yaacob et al., 2014, and De Jager et al., 2017) confirmed this tendency towards a preference for powered toothbrushes, which resulted in being more efficient, convenient, and easier for patients to learn to use [14,15].

Nowadays, the research on this field has shifted its aim to the comparison of different types of powered toothbrushes on the market, such as toothbrushes with oscillating/rotating heads, and toothbrushes with sonic action heads. Even though there are a lot of clinical and in vitro studies on this topic, there is currently only one review analysing this comparison [16]. The review of Deacon et al. (2010) on 17 trials with a total of 1369 participants reported that rotating/oscillating toothbrushes reduce plaque and gingivitis more than side-to-side brushes (sonic) in the short term, but this difference is small, and the clinical relevance is unclear [17]. This review included different types of powered toothbrushes, while the aim of the present narrative review was to compare the effects on oral hygiene of the two most used types of power toothbrushes by patients: oscillating rotating heads (ORH) vs sonic action heads (SAH) toothbrushes to maintain gingival, periodontal, and peri-implant health.

## 2. Materials and Methods

This review protocol followed the PRISMA guidelines.

The outcome measures evaluated were plaque accumulation and gingival inflammation, measured as bleeding on probing.

### 2.1. Focused Question (PICO)

The focused question used was: ‘Do sonic toothbrushes vs oscillating/rotating toothbrushes have an additional effect on the reduction of plaque and gingival inflammation of patients’?

### 2.2. Study Selection and Eligibility Criteria

Randomised Clinical Trials (RCTs) comparing the efficacy of oral hygiene with an electric oscillating (ORH) or electric sonic (SAH) heads toothbrush were searched. Studies with the following inclusion criteria were selected: (i) patients aged between 15 to 70 years, (ii) patients without severe systemic or psychiatric disease, and (iii) the parameters studied were: plaque accumulation and bleeding on probing (as an outcome of gingival inflammation). The gingival recession was evaluated as a secondary outcome of interest in order to compare the possible association of one type of electric toothbrush with gingival recession prevalence. Reviews, letters, and vitro studies were excluded. Other exclusion criteria were orthodontics patients, patients with disabilities, studies comparing two or more different types of sonic toothbrushes, studies comparing two or more different types of oscillating/rotating toothbrushes, final timepoint after less than 15 days, studied year of publication before 2000.

### 2.3. Search Strategy

The search was carried out independently by one authors (C.P.) and confirmed by another author (S.P.) on five databases (MEDLINE, Embase, Inspec, PQ SciTech and BIOSIS) using synonyms as “sonic” OR “powered” OR “electric” OR “rota” OR “rotating” OR “rotation” OR “oscillated” OR “oscillating” OR “ultrasonic” AND “toothbrush” with different combinations. The search was limited to articles in English. Regarding publication date or follow-up no restrictions were applied when searching the electronic databases to be as inclusive as possible. The exclusion criteria were applied after this electronic searching. The bibliographies of all included studies were checked in order to identify other possible RCTs related to the topic. A complementary manual search which included a complete revision up to April 2018 was made of the following journals: Journal of Oral Science & Rehabilitation and European Journal of Inflammation.

### 2.4. Screening and Selection of Papers

Titles identified through the search strategy were uploaded on the program End-Note (ISI Research software 2001, Berkeley, CA, USA) to exclude duplicates.

Plus, titles and abstracts of all remaining articles were independently scanned by three reviewers (F.E., S.P. and E.P.), followed by inclusion and exclusion criteria. Disagreements between the authors were solved by a discussion with one other author (S.C.). Full-text was downloaded of the studies appearing to meet the inclusion criteria or with insufficient information in the abstract. Two reviewers (F.E. and S.P.) read independently the full-test articles to decide if the studies could be included in the present review. Disagreements were solved by a discussion between the two authors. In case a resolution was not easy, a third reviewer (E.P.) was consulted. Studies rejected from this point on were recorded in a table of excluded studies, explaining reasons for their exclusion. All full-text articles meeting the inclusion criteria were evaluated again by the four authors (S.C., F.E., S.P., and E.P.) to assess the quality assessment of each article and to extract data of clinical parameters.

### 2.5. Risk of Bias in Individual Studies

The risk of bias assessment was performed by reviewers independently by a process of quality analysis, following the Cochrane Reviewers’ guidelines. All disagreements were resolved after discussion. A ‘funnel plot’ was used for all studies included.

### 2.6. Data Analysis and Statistics

The Plaque Index (PI) reduction, and the reduction of the percentage of sites with Bleeding on Probing (BoP), were expressed as the average difference between baseline and follow-up, after the use of each toothbrush.

In these RTCs, parameters for plaque and gingival inflammation were reported through different clinical indices. To perform a statistical comparison between these articles, outcomes were converted and summarized before using the random-effects models. Differences between (ORHs) and (SAHs) were expressed as weighted mean differences (WMD) and a 95% confidence interval (CI). A *p*-value of statistical significance was defined <0.5 as an indicator of heterogeneity and data were considered heterogeneous for value higher than 40%. Mean differences and standard errors were entered for each study. When data were not expressed in terms of mean differences, the mean difference was calculated as well as an estimation of the standard deviation and percentage.

## 3. Results

The combined search among different databases provided 540 records (Figure 1). After removing the duplicates using the software EndNote (ISI Research software 2001), 397 titles were identified. After the screening of each title and abstract of these 397 references, according to the relevance of the topic, 113 articles remained. Following inclusion and exclusion criteria, the full text of these articles was obtained for 35 studies. Only 18 papers appeared to follow inclusion criteria, and after a last revision of the authors, 12 articles were included in the analysis. The 23 articles’ full texts were excluded for different reasons, resumed in Table 1 [18,19,20,21,22,23,24,25,26,27,28,29,30,31,32,33,34,35,36,37,38,39,40].

The principal characteristic of the 12 included studies were resumed in Table 2 [41,42,43,44,45,46,47,48,49,50,51,52]. A total of 1433 patients were analysed, comparing the domiciliary use of an oscillating/rotating toothbrush, or a sonic one, after a follow-up variable for a minimum of four weeks, up to a maximum of 24 weeks (mean: 7.83 weeks).

In Table 3, the main conclusions of each study are reported. In six studies, oscillating/rotating toothbrushes showed better clinical results compared to sonic toothbrushes. Nevertheless, four of these studies belong to the same research group and, moreover, in only one study the statistical significance (*p* < 0.05) is clearly reported. All of these six studies were sponsored by Procter & Gamble, which supports the company Oral-B, producing ORH toothbrushes. One study reported better clinical outcomes in patients using ORH toothbrushes, but with no statistical differences and no conflict of interest (Table 4). In one study, the ORH group had significantly better results than the SAH group in the mean Plaque Index at four and 12 weeks (*p* < 0.05), while SAH was significantly greater in the Gingival Index at four weeks. Therefore, both automatic toothbrushes had similar clinical outcomes in this study.

In the other four studies, patients who used sonic toothbrushes had better clinical outcomes, but only in three of them the differences were statistically significant. It must be considered that one of them was the study of Ricci et al., (2014), that had a too short follow-up (two weeks), and the other one was the study of Starke et al., (2017), which was funded by Philips [47,48,49,50].

The quality and bias of the study are reported in Table 5.

The findings about the GI, BoP, and PI were compared in all the studies. Sometimes, different parameters were used to analyse these aspects in these clinical studies, so that in order to compare the clinical results of the oscillating toothbrushes and sonic toothbrushes, the percentages of change from each baseline were calculated. In Figure 2, Figure 3 and Figure 4, respectively, the distribution of the mean Gingival Index, Bleeding Score, and Plaque Index for the sonic toothbrushes (first line/dark) and oscillating ones (second line/clear) were reported. The mean values of these clinical parameters (changes) were weighted considering the number of implants of each study, so that differences between two intervention groups were calculated with a significance < 0.05.

## 4. Discussion

The experimental gingivitis studies from the 1960s produced the common accepted concept that bacterial plaque is essential to the initiation of gingivitis and, if unresolved, would lead to periodontitis [54].

The most widespread mechanical means of controlling plaque at home is the toothbrush, preferably used in combination with a fluoride toothpaste (according to the American Dental Association). In addition, the ideal brushing technique appears to be the one that allows for complete plaque removal in the least possible time, without causing any damage to tissue [55].

In the practice of evidence-based dentistry, every dental professional must make a considered decision regarding his or her advice to each patient. The method of collecting information for a systematic review provides a solid base for clinical decision-making. [56] The present review was aimed at comparing the effects of two different electric toothbrushes on plaque accumulation and the Gingival Index. In this regard, according to the literature, there is statistically significant evidence that, compared with manual toothbrushes, powered toothbrushes are more effective in reducing plaque and gingivitis in long- and short-term studies [14]. On the contrary, ORHs toothbrushes and side-to-side action brushes were not found to differ from each other for either plaque or gingivitis reduction in the long term (>3 months) [17].

The studies selected for this review had diverse study designs with follow-ups, ranging from two weeks to 24 weeks. The toothbrushes used and the outcomes measured also varied in each study. Therefore, the clinical results of both oscillating toothbrushes and sonic toothbrushes were compared throughout the percentages of changing from the baseline, respectively corresponding to the clear (ORHs) and dark (SHAs) line.

In two of the selected studies, significant differences between ORHs and SAHs toothbrushes were not found for any of the clinical parameters evaluated [41,52].

Great improvements in the Gingival Index were demonstrated for both ORHs and SAHs toothbrushes (Figure 2). Five of the included articles showed that ORHs moderate statistically reduced the Gingival Index, compared to the SAHs [42,43,44,45,51].

However, all of these studies resulted in being supported by the company Oral-B, and three of them were carried out by the same authors; moreover, statistics and p-values were not clear, and data were reported incompletely. Otherwise, results from three of the selected studies, one of which was funded by Philips, showed significant Gingival Index improvements with sonic toothbrushes, when compared to rotating oscillating toothbrushes [46,49,50].

Schmickler et al. reported that SAH-assigned subjects experienced a 20.34% (23.73–3.19%) Gingival Index mean reduction (dark line) versus the 8.2% (19.18–10.98%) obtained with ORH toothbrushes (clear line). Besides, it is to be noted that this was a long-term follow up study (12 weeks), and it was qualified as a low-risk research.

Based on the provided data on the reduction of plaque (Figure 4), two of the included articles showed significant differences in favor of the ORHs toothbrush: In William et al.’s article, data were not completely reported, and in Patter et al.’s article, despite the benefits shown on the plaque score, it is worth noting that the Gingival Index had significant improvements in the SAH toothbrush group. In one study, more significant results in terms of the Plaque Index were observed in favor of SAHs, but the data are incomplete.

Concerning the bleeding score (Figure 3), ORHs electric toothbrushes were found to provide significant reductions in four studies out of 12. However, all of these articles belong to the same research group, and the p-value was not clearly defined [42,43,44,45]. Contextually, SAHs demonstrated to be statistically superior compared to the ORHs toothbrushes in three of the selected trials, although the Starke et al. one was funded by Philips, and Ricci et al.’s one does not report complete data [47,49,50].

Nevertheless, Schmickler et al.’s investigation concluded that when the brush head is not replaced after a period of four months, an SAH toothbrush delivers a significant bleeding score reduction within 16 weeks of follow-up. This is contrary to the ORH toothbrush, which did not induce significant improvements at any follow-up.

Finally, on analysing the assessment of the quality of the 12 articles (Table 5), it emerges that the overall risk of bias for the included trials was, in general, medium low. However, only three of them can be defined as good quality studies, two of which resulted in being in favour of the SAHs toothbrush [28,29,52].

Interpreting all these findings, one can suggest that no great differences were observed between ORHs and SAHs toothbrushes. Both provide great improvements in plaque removal and gingival inflammation reduction. Furthermore, none among the 12 studies showed evidence on ORH or SAHs toothbrush damage to soft or hard tissue. Regarding this aspect, it would be plausible to aspect more tissue damage in ORH. Considering ORHs functioning principles, it is to be noted that the strength vectors of the force of rotation could be perpendicular to the gingival sulcus bringing mechanical stress. It might be interesting to investigate whether these strength vectors are able to push bacteria under the gingival sulcus.

In the authors’ opinion, patients should be educated on domiciliary oral hygiene. It is important to explain the fundamental topics of periodontal and peri-implant diseases etiology in order to gain patients’ attention and compliance: nobody likes things they do not understand. Patients have to learn from the dentist and the dental hygienist new concepts in dentistry such as microbiome and systemic correlations with local inflammations. Sonic and electric toothbrushes could be a simple way to standardize oral home care, focusing the “motivation session” on the etiology of pathology and the important of prevention rather than spending time to show manual toothbrush techniques. It has been demonstrated by modern literature that manual toothbrushes could even damage the gingival tissue and they have too many patients-related biases compared to automatic toothbrushes.

## 5. Conclusions

In conclusion, the combined evidence presented in included RCTs showed that the use of both electric toothbrushes results in being greater than manual toothbrushes in managing oral hygiene, without causing damage to soft or hard tissue, but it is difficult to draw concrete conclusions in relation to the efficacy of a specific type of electric toothbrush. Nevertheless, according to the quality assessment of the 12 articles, two of the three investigations were defined as good quality studies, resulting in being in favor of SAHs toothbrushes, and, in line with the provided clinical outcomes, greater long-term results were shown in favor of SAHs toothbrushes for both the gingival and bleeding indices. Therefore, it could be concluded that better improvement tendencies in clinical parameters were observed in SAHs electric toothbrushes, but further good quality trials are necessary in order to outline more accurate conclusions.

## Figures and Tables

**Figure 1 ijerph-18-01468-f001:**
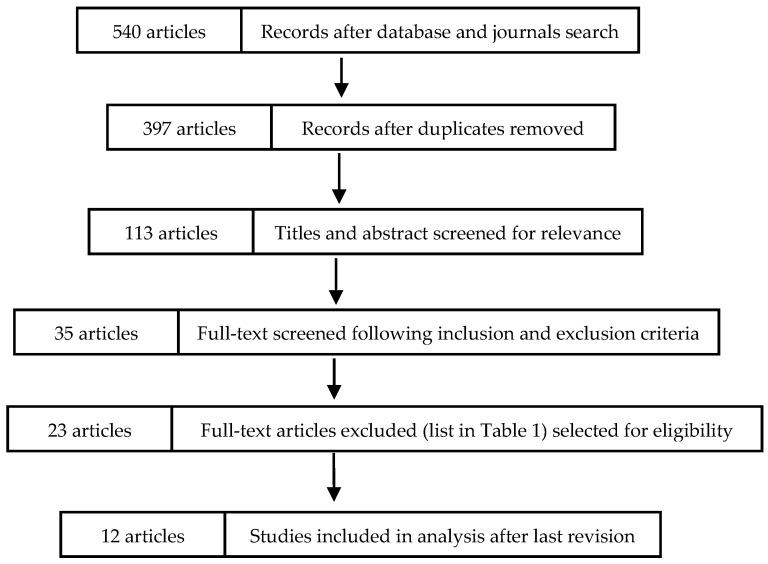
Articles inclusion. Flow chart diagram (2009) of search strategy adapted from PRISMA.

**Figure 2 ijerph-18-01468-f002:**
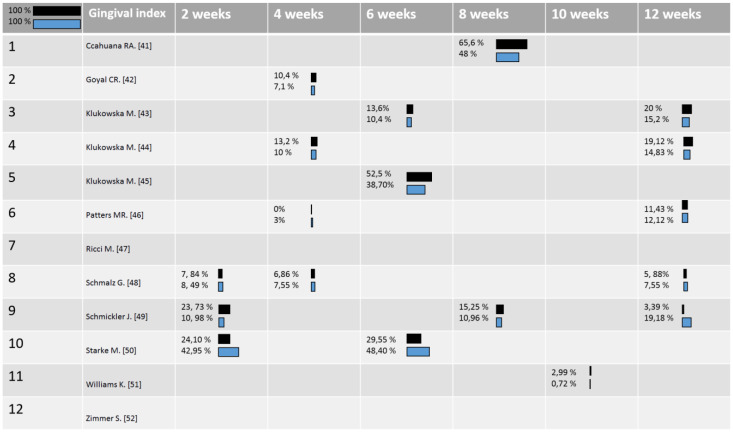
Gingival Index, the mean values of Gingival Index (changes) in relation to the number of patients of each study at each time-point: blue color correspond to ORHs and black corresponds to SAHs toothbrushes [41,42,43,44,45,46,47,48,49,50,51,52].

**Figure 3 ijerph-18-01468-f003:**
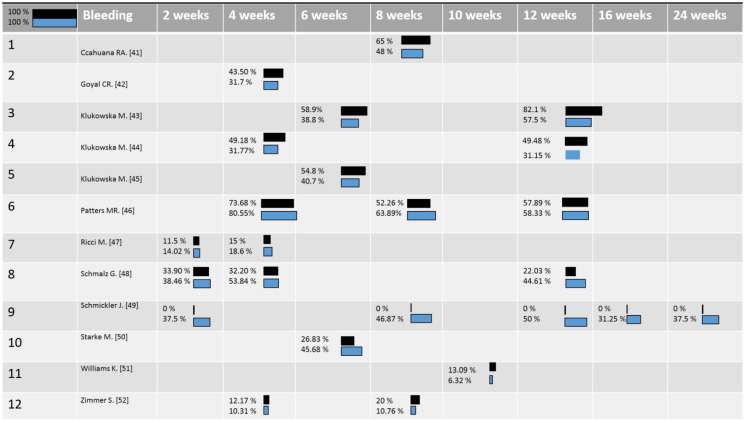
Bleeding Index, the mean values of Bleeding Index (changes) in relation to the number of patients in each study at each time-point: blue color corresponds to ORHs and black to SAHs toothbrushes [41,42,43,44,45,46,47,48,49,50,51,52].

**Figure 4 ijerph-18-01468-f004:**
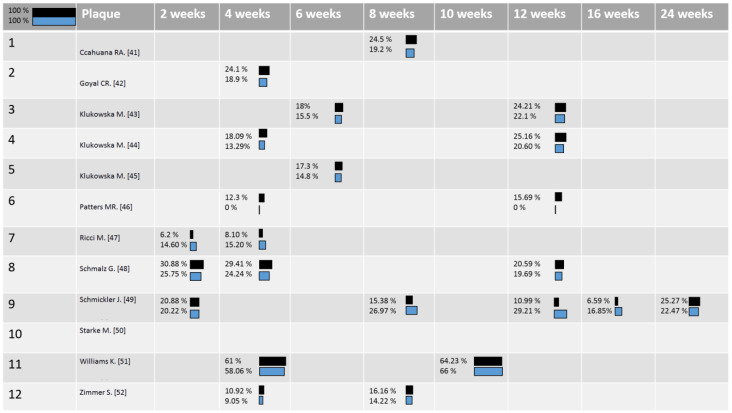
Plaque Index, the mean values of Plaque Index (changes) in relation to the number of patients in each study at each time-point: blue color corresponds to ORHs and black to SAHs toothbrushes [41,42,43,44,45,46,47,48,49,50,51,52].

**Table 1 ijerph-18-01468-t001:** Excluded articles after full-text screening.

Nr.	References	Exclusion Motivation
1	Bader HI, Boyd RL. Comparative efficacy of a rotary and a sonic powered toothbrush on improving gingival health in treated adult periodontitis patients. Am J Dent. 1999 Jun;12(3):143-7. [18]	Year of publication
2	Biesbrock AR, Bartizek RD, Gerlach RW, Terézhalmy GT. Oral hygiene regimens, plaque control, and gingival health: a two-month clinical trial with antimicrobial agents. J Clin Dent. 2007;18(4):101-5. [19]	Short follow-up: single brushing
3	Biesbrock AR, He T, Walters PA, Bartizek RD. Clinical evaluation of the effects of a sonic toothbrush with ultrasound waveguide in disrupting plaque with and without bristle contact. Am J Dent. 2008 Apr;21(2):83-7. [20]	Comparison between the same sonic toothbrush with the power turned on vs turned off
4	Biesbrock AR, Walters PA, Bartizek RD, Goyal CR, Qaqish JG. Plaque removal efficacy of an advanced rotation-oscillation power toothbrush versus a new sonic toothbrush. Am J Dent. 2008 Jun;21(3):185-8. [21]	Short follow-up
5	Costa MR, Silva VC, Miqui MN, Sakima T, Spolidorio DM, Cirelli JA. Efficacy of ultrasonic, electric and manual toothbrushes in patients with fixed orthodontic appliances. Angle Orthod. 2007 Mar;77(2):361-6. doi: 10.2319/0003-3219(2007)077[0361:EOUEAM]2.0.CO [22]; 2.	Orthodontic patients aged between 12 and 18 years old
6	Goyal CR, Qaqish J, He T, Grender J, Walters P, Biesbrock AR. A randomized 12-week study to compare the gingivitis and plaque reduction benefits of a rotation-oscillation power toothbrush and a sonic power toothbrush. J Clin Dent. 2009;20(3):93-8. [23]	Data appeared to duplicate another study
7	Goyal CR, Qaqish JG, Sharma NC, Warren PR, Cugini M, Thompson MC. Plaque removal efficacy of a novel tooth wipe. J Clin Dent. 2005;16(2):44-6. [24]	Short follow-up: single brushing
8	Goyal CR, Sharma NC, Qaqish JG, Cugini MA, Thompson MC, Warren PR. Efficacy of a novel brush head in the comparison of two power toothbrushes on removal of plaque and naturally occurring extrinsic stain. J Dent. 2005 Jun;33S1:37-43. [25]	Not following all inclusion criteria because it is focused on dental stains and not on plaque or gingival inflammation
9	Grossman E, Dembling W, Proskin HM. A comparative clinical investigation of the safety and efficacy of an oscillating/rotating electric toothbrush and a sonic toothbrush. J Clin Dent. 1995;6(1):108-12. [26]	Year of publication
10	Hanato Y, Kishimoto T, Ojima M, Matsuo T, Kanesaki N, Ryu C, Hanioka T. Comparative study of plaque removal efficacy of twin-motor sonic toothbrush with floating bristles and conventional powered toothbrushes in posterior teeth. Am J Dent. 2005 Aug;18(4):237-40. [27]	Short follow-up: single brushing
11	Williams K, Haun J, Dockter K, Ferrante A, Bartizek RD, Biesbrock AR. A plaque removal study comparing two advanced-design oscillating battery-powered toothbrushes. J Clin Dent. 2003;14(4):88-92. [28]	Short follow-up: single brushing
12	Hefti AF, Stone C. Power toothbrushes, gender, and dentin hypersensitivity. Clin Oral Investig. 2000 Jun;4(2):91-7. doi: 10.1007/s007840050122. [29]	Not following all inclusion criteria because it is focused on hypersensitivity and not on plaque or gingival inflammation
13	Klukowska M, Grender JM, Goyal CR, Qaqish J, Biesbrock AR. 8-week evaluation of anti-plaque and anti-gingivitis benefits of a unique multi-directional power toothbrush versus a sonic control toothbrush. Am J Dent. 2012 Sep;25 Spec No A(A):27A-32A. [30]	First step, after 8 weeks evaluation included the same study of 12 weeks
14	Klukowska M, Grender JM, Conde E, Ccahuana-Vasquez RA, Goyal CR. A randomized 12-week clinical comparison of an oscillating-rotating toothbrush to a new sonic brush in the reduction of gingivitis and plaque. J Clin Dent. 2014;25(2):26-31. [31]	Data appeared to duplicate another study
15	Klukowska M, Grender JM, Conde E, Goyal CR, Qaqish J. A six-week clinical evaluation of the plaque and gingivitis efficacy of an oscillating-rotating power toothbrush with a novel brush head utilizing angled CrissCross bristles versus a sonic toothbrush. J Clin Dent. 2014;25(2):6-12. [32]	First step, after 6 weeks evaluation, included study of 12 weeks
16	Robinson PJ, Maddalozzo D, Breslin S. A six-month clinical comparison of the efficacy of the Sonicare and the Braun Oral-B electric toothbrushes on improving periodontal health in adult periodontitis patients. J Clin Dent. 1997;8(1 Spec No):4-9. [33]	Year of publication
17	Sharma NC, Galustians J, Qaqish J, Cugini M. A comparison of two electric toothbrushes with respect to plaque removal and subject preference. Am J Dent. 1998 Sep;11(Spec No):S29-33. [34]	Short follow-up
18	Sharma NC, Lyle DM, Qaqish JG, Galustians J. Evaluation of the plaque removal efficacy of three power toothbrushes. J Int Acad Periodontol. 2006 Jul;8(3):83-8. [35]	Short follow-up: single brushing
19	Singh G, Mehta DS, Chopra S, Khatri M. Comparison of sonic and ionic toothbrush in reduction in plaque and gingivitis. J Indian Soc Periodontol. 2011 Jul;15(3):210-4. doi: 10.4103/0972-124X.85662. [36]	It is focused on the comparison of sonic toothbrush with ionic toothbrush
20	Thienpont V, Dermaut LR, Van Maele G. Comparative study of 2 electric and 2 manual toothbrushes in patients with fixed orthodontic appliances. Am J Orthod Dentofacial Orthop. 2001 Oct;120(4):353-60. doi: 10.1067/mod.2001.116402. [37]	Orthodontic patients
22	Van der Weijden GA, Timmerman MF, Reijerse E, Snoek CM, Van der Velden U. Comparison of an oscillating/rotating electric toothbrush and a ‘sonic’ toothbrush in plaque-removing ability. A professional toothbrushing and supervised brushing study. J Clin Periodontol. 1996 Apr;23(4):407-11. doi: 10.1111/j.1600-051x.1996.tb00565.x. [38]	Short follow-up
22	Williams KB, Cobb CM, Taylor HJ, Brown AR, Bray KK. Effect of sonic and mechanical toothbrushes on subgingival microbial flora: a comparative in vivo scanning electron microscopy study of 8 subjects. Quintessence Int. 2001 Feb;32(2):147-54. [39]	Not following all inclusion criteria because it is focused on subgingival microbial and not on plaque or gingival inflammation
23	Williams K, Rapley K, Huan J, Walters P, He T, Grender J, Biesbrock AR. A study comparing the plaque removal efficacy of an advanced rotation-oscillation power toothbrush to a new sonic toothbrush. J Clin Dent. 2008;19(4):154-8. [40]	Short follow-up and not following all inclusion criteria

**Table 2 ijerph-18-01468-t002:** All studies included for the qualitative analysis.

Nr.	References	Patients	Age Range of Patients	SAH	ORH	Plaque Index	Gingival Index	Follow-Up Period And Frequency Use
1	Ccahuana-Vasquez RA. [41]	148 = 75 (OR) + 73 (SA)	>18 years old—43.9 (SD = 11.30)	Sonicare DiamondClean	Oral-B Professional Care 1000	Rustogi Modified Navy Plaque Index (RMNPI)	Modified Gingival Index (MGI) Gingival Bleeding Index (GBI)	8 weeks Twice/day
2	Goyal CR. [42]	130 = 65 (OR) + 65 (SA)	>18 years old—42.1 (SD = 11.20)	Philips Sonicare Essence 5500	Oral-B Professional Deep Sweep	Rustogi Modified Navy Plaque Index (RMNPI)	Modified Gingival Index (MGI) Gingival Bleeding Index (GBI)	4 weeks Twice/day
3	Klukowska M. [43]	130 = 65 (OR) + 65 (SA)	>18 years old—44.7	Sonicare DiamondClean	Oral-B Triumph with SmartGuide	Rustogi Modified Navy Plaque Index (RMNPI)	Modified Gingival Index (MGI) Gingival Bleeding Index (GBI)	12 weeks Twice/day
4	Klukowska M. [44]	127 = 62 (OR) + 65 (SA)	>18 years old—36.2 (12.02)	Colgate ProClinical A1500	Oral-B Triumph with SmartGuide	Rustogi Modified Navy Plaque Index (RMNPI)	Modified Gingival Index (MGI) Gingival Bleeding Index (GBI)	12 weeks Twice/day
5	Klukowska M. [45]	130 = 65 (OR) + 65 (SA)	>18 years old—36.2 (12.87)	Colgate ProClinical A1500	Oral-B Pro 7000 SmartSeries with SmartGuide with Oral-B CrossAction brush head, D34/EB50	Rustogi Modified Navy Plaque Index (RMNPI)	Modified Gingival Index (MGI) Gingival Bleeding Index (GBI)	6 weeks Twice/day
6	Patters MR. [46]	70 = 35 (OR) + 35 (SA)	>18 years old	Sonicare PLUS	Braun Oral-B 3D Excel, model D17525	Plaque Index (PI)	Modified Gingival Index (MGI)	12 weeks Twice/day
7	Ricci M. [47]	30 = 15 (OR) + 15 (SA)	>18 years old—35.8 (10.02)	Philips Sonicare	Oral B power toothbrush	Plaque Index (PI)	Full Mouth Bleeding Score (FMBS)	4 weeks Twice/day
8	Schmalz G. [48]	25 = 12 (OR) + 13 (SA)	>18 years old—28.5 (10.2)	Philips Sonicare	Oral B ProfessionalCare 7000	Quigley-Hein plaque index (QHI)	Gingival Index (GI) Papilla Bleeding Index (PBI),	12 weeks Twice/day
9	Schmickler J. [49]	284 = 142 (OR) + 142 (SA)	>18years old—65	Differed types of oscillating/ rotating toothbrushes	Differed types of sonic action toothbrushes	Quigley-Hein plaque index (QHI),	Papilla Bleeding Index (PBI), and Gingival Index (GI)	24 weeks Twice/day
10	Starke M. [50]	179 = 90 (OR) + 89 (SA)	>18 years old—42.2 (12.2)	Philips Sonicare DiamondClean	Oral-B 7000	Modified Plaque Index (MPI)	Gingival Bleeding Index (GBI)	6 weeks Twice/day
11	Williams K. [51]	80 = 40 (OR) + 40 (SA)	>18 years old—34.1 (12.3)	Sonicare FlexCare	Oral-B Triumph	Turesky Modification of the Quigley-Hein Plaque Index	Löe-Silness Gingivitis Index	10 weeks Twice/day
12	Zimmer S. [52]	100 = 50 (OR)+ 50 (SA)	18 > years old < 30	Philips SoniCare	Oral-B^®^ Professional Care 7000	Turesky Modification of the Quigley-Hein Plaque Index	Gingival Index (GI) Loë-Silness Papillary Bleeding Index	12 weeks Twice/day

SAH: Sonic Action Heads; ORH: Rotating-Oscillating Heads; OR: Rotating-Oscillating; SA: Sonic Action; SD: Standard Deviation.

**Table 3 ijerph-18-01468-t003:** Main conclusion of each included study with significance and conclusions.

Nr.	References	Better Clinical Results in:	Significance (*p* < 0.05)	Conclusion
1	Ccahuana-Vasquez RA. [41]	OR	No	Both types of toothbrush had significant results on clinical parameters compared to baseline (*p* < 0.001) but no significant differences were reported between OR and SA.
2	Goyal CR. [42]	OR	YES	Both types of toothbrush had significant results on clinical parameters compared to baseline (*p* < 0.001). Moreover, it is declared in the results that OR significantly reduced Gingival Index and bleeding, 40—50 % greater, compared to the SA, nevertheless the statics and *p*-value is not clear.*
3	Klukowska M. [43]	OR	YES	Both types of toothbrush had significant results on clinical parameters compared to baseline (*p* < 0.001). Moreover, it is declared in the results that OR reduced Gingival Index and bleeding significantly greater compared to the SA, but the statics and *p*-value is not clear. *
4	Klukowska M. [44]	OR	YES	Both types of toothbrush had significant results on clinical parameters compared to baseline (*p* < 0.001). Moreover, it is declared in the results that OR reduced Gingival Index and bleeding significantly greater compared to the SA, but the statics and p-value is not clear. *
5	Klukowska M. [45]	OR	YES	Both types of toothbrush had significant results on clinical parameters compared to baseline (*p* < 0.001). Moreover, it is declared in the results that OR reduced Gingival Index and bleeding significantly greater compared to the SA, but the statics and p-value is not clear. *
6	Patters MR. [46]	/	/	The OR group differed significantly from the SA group in mean Plaque Index at 4 and 12 weeks (*p* < 0.05) while SA was significantly greater in Gingival Index at 4 weeks. In total, OR had better but not significant clinical outcomes.
7	Ricci M. [47]	SA	YES	Both types of toothbrush had significant results on clinical parameters compared to baseline (*p* < 0.001). Moreover, it is declared in the results that SA significantly reduced Plaque Index and bleeding score compared to the OR, with a *p* value < 0.05, but data are not reported completely in the study.
8	Schmalz G. [48]	SA	/	Both types of toothbrush had significant results on clinical parameters compared to baseline (*p* < 0.05). Moreover, it is declared that significant differences were found for same parameters, although the changes were just minor. SA groups shown a constant decrease in Gingival Index, plaque and bleeding score while in the OR group the decrease was major at the beginning of observation period.
9	Schmickler J. [49]	SA	YES	The SA toothbrush had a significant better influence on Gingival Index and bleeding over OR group if the toothbrush is not replaced after a period of 4 months. Both type of toothbrush head loss the effectiveness in removing plaque after a period of 16—24 weeks.
10	Starke M. [50]	SA	YES	SA toothbrush was statistically superior to the OR in reducing gingival inflammation, gingival bleeding, after a timing of 14 and 42 days of home use.
11	Williams K. [51]	OR	YES	OR toothbrush had significant efficient results in reducing plaque and gingival inflammation but the data are reported incompletely.
12	Zimmer S. [52]	OR	NO	OR and SA toothbrush had significant better results than manual toothbrush in reducing plaque and bleeding but differences between the two powered toothbrushes was not significative.

* = In clinical study, especially in dentistry and oral hygiene, to indicate the common outcome, very small numbers are often used, for example 3 mm of probing depth, 1 mm of marginal bone loss, etc. Example of a very common result case: Treatment 1 had a mean gain of 0.4 mm and Treatment 2 had a mean gain of 0.2 mm but together with this value, the standard deviation is visible to indicate to the reader the variation in the sample. T-student is a simple statistic test used to compare two variables; it considers all mean values of the two groups to indicate if the results, despite very small numerical value, are significant or not. If only percentage is used between the two values, Treatment 1 has double the benefits of Treatment 2. It is acceptable and recommended to calculate the percentage of Treatment 1 and 2 compared to the baseline outcome, but it is not correct to calculate the percentage of Treatment 2 to check the significance of the results.

**Table 4 ijerph-18-01468-t004:** Conflicts of interest for each included study.

Nr.	References	Conclusion
1	Ccahuana-Vasquez RA. [41]	4 authors out of 6 (Dr. Ccahuana-Vasquez, Dr. Conde, Dr. Grender and Ms. Cunningham are employees of Procter & Gamble that supported the study and own the company Oral-B). Dr. Goyal and Mr. Qaqish have no conflict.
2	Goyal CR. [42]	3 authors out of 5 (Dr. Klukowska, Dr. Grender, and Ms. Cunningham are employees of Procter & Gamble that supported the study and own the company Oral-B). Dr. Goyal and Mr. Qaqish have no conflict.
3	Klukowska M. [43]	4 authors out of 5 (Dr. Klukowska, Dr. Grender, Biesbrock, and Mandl are employees of Procter & Gamble that supported the study and own the company Oral-B). Dr. Goyal has no conflict.
4	Klukowska M. [44]	3 authors out of 4 (Dr. Klukowska, Dr. Grender, and Conde are employees of Procter & Gamble that supported the study and own the company Oral-B). Dr. Goyal has no conflict.
5	Klukowska M. [45]	4 authors out of 5 (Dr. Klukowska, Dr. Grender, Ms. Conde, and Dr. Ccahuana-Vasquez are employees of Procter & Gamble that supported the study and own the company Oral-B). Dr. Goyal has no conflict.
6	Patters MR. [46]	This study was funded by a grant for Oralbotic Research, Inc. (company relative to the third group of treatment: powered toothbrush Hydrabrush^®^)
7	Ricci M. [47]	No conflict of interest
8	Schmalz G. [48]	No conflict of interest
9	Schmickler J. [49]	No conflict of interest
10	Starke M. [50]	This study was funded by Philips
11	Williams K. [51]	4 authors out of 7 (Walters, Tao He, Julie Grender, and Biesbrock are employees of Procter & Gamble that supported the study and own the company Oral-B). Dr. Williams, Rapley, and Haun have no conflict.
12	Zimmer S. [52]	No conflict

**Table 5 ijerph-18-01468-t005:** Quality assessment: The methodological quality was assessed using the colors green, yellow, and red if the study is respectively with low risk, fair quality, or high risk, as suggested by the Cochrane Handbook in 2011 [53].

		Quality Assessment							
Nr.	References	Selection Generation	Allocation Concealement	Blinding of Participants and Personnel	Blinding of Outcome Assessment	Incomplete Outcome Data	Selective Reporting	Other Bias	Study Quality
1	Ccahuana-Vasquez RA. [41]	Low Risk	Low Risk	Low Risk	Low Risk	Low Risk	Low Risk	High Risk	Fair Quality
2	Goyal CR. [42]	Low Risk	Low Risk	Low Risk	Low Risk	Low Risk	Low Risk	High Risk	Fair Quality
3	Klukowska M. [43]	Low Risk	Low Risk	Low Risk	Unclear Risk	Low Risk	Low Risk	High Risk	Fair Quality
4	Klukowska M. [44]	Low Risk	Low Risk	Low Risk	Unclear Risk	Low Risk	Low Risk	High Risk	Fair Quality
5	Klukowska M. [45]	Low Risk	Low Risk	Low Risk	Low Risk	Low Risk	Low Risk	High Risk	Fair Quality
6	Patters MR. [46]	Low Risk	Low Risk	Low Risk	Unclear Risk	Unclear Risk	High Risk	High Risk	Poor quality
7	Ricci M. [47]	Unclear Risk	Low Risk	Low Risk	Unclear Risk	Unclear Risk	High Risk	Unclear Risk	Poor quality
8	Schmalz G. [48]	Low Risk	Low Risk	Low Risk	Unclear Risk	Low Risk	Low Risk	Low Risk	Good quality
9	Schmickler J. [49]	Low Risk	Low Risk	Low Risk	Unclear Risk	Low Risk	Low Risk	Low Risk	Good quality
10	Starke M. [50]	Low Risk	Low Risk	Low Risk	Unclear Risk	High Risk	High Risk	High Risk	Poor quality
11	Williams K. [51]	Low Risk	Low Risk	Low Risk	Low Risk	Low Risk	Low Risk	High Risk	Fair Quality
12	Zimmer S. [52]	Low Risk	Low Risk	Low Risk	Low Risk	Low Risk	Low Risk	Low Risk	Good quality

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
