# Peer review of "The Efficacy of Powered Oscillating Heads vs. Powered Sonic Action Heads Toothbrushes to Maintain Periodontal and Peri-Implant Health: A Narrative Review"

_ijerph, 2021, doi:10.3390/ijerph18041468_

Round 1
Reviewer 1 Report
Dear Authors,
the paper is interesting, but in order to be improved some revisions are requested. Major drawbacks are presented below by the following points:
- Page 1 line 16: Inform what is abbreviation RCT?
- Page 1 line 39, 44, etc: Check citations format throughout the paper. In the text, reference numbers should be placed in square brackets [ ], and placed before the punctuation; for example [1], [1–3] or [1,3]. For embedded citations in the text with pagination, use both parentheses and brackets to indicate the reference number and page numbers; for example, [5] (p. 10). or [6] (pp. 101–105). Please consult: https://www.mdpi.com/journal/ijerph/instructions#preparation
- Page 2 line 77 to 79: „The aim of the present review was to compare the effects on oral hygiene of two 77 different types of power toothbrushes: oscillating/rotating heads (ORHs) vs sonic action heads (SAHs).“ Informed in introduction section.
- Page 3 line 131: Inform what is abbreviations PI and BoP?
- Page 3 line 139: Clarify abbreviations Q and I2.
- Page 3 line 143: Explain the formula.
- Page 13 line 231: It is unclear what the number 14 means: “…long and short term14.“
- Page 15 to 17 line 319 to 448: Check citations format references in references section.
Author Response
Dear reviewer, thank you for your precious suggestions. It was useful to improve the manuscript.
The abbreviation RCT is explained in the introduction, not in the abstract, pag 2, line 86.
Checked the citations format throughout the paper.
Page 2 line 77 to 79: I deleted the sentence because it was already explained in the end of the introduction “The aim of the present review was to compare the effects on oral hygiene of two 77 different types of power toothbrushes: oscillating/rotating heads (ORHs) vs sonic action heads (SAHs).“
Page 3 line 131: We informed the abbreviations PI and BoP.
Pag 3: We simplify the statistics.
Pag 13: Yes, it was an error. It is: long and short term studies [14].
Page 15 to 17 line 319 to 448: I agree with reviewer and we improved the manuscript according to it.

Reviewer 2 Report
This manuscript is a narrative review with the aim to compare the efficacy of rotating-oscillating heads (ORHs) VS sonic action heads (SAHs) powered toothbrushes on oral health maintenance, considering plaque accumulation and gingival inflammation as outcome2. The importance of the topic for this review is alluded to but needs to be better justified, and some considerations need to be addressed.
To guide this evaluation, this reviewer used SANRA, the Scale for the Assessment of Narrative Review Articles (Baethge, C., Goldbeck-Wood, S. & Mertens, S. SANRA—a scale for the quality assessment of narrative review articles. Res Integr Peer Rev 4, 5 (2019). https://doi.org/10.1186/s41073-019-0064-8). According to this scale, the manuscript would receive 10 points out of 12, indicating that it is a narrative with very good quality.
- Introduction:
Many small paragraphs with single sentences. Try to combine them into 3 to 4 paragraphs.
The title of the article states that the evaluation will be in relation to Periodontal and Peri-implant Health maintenance. However, this topic is not addressed in the introduction. Nor it is included in the article’s aim, which to compare the effects on oral hygiene of the two most used types of power toothbrushes by patients: oscillating rotating heads (ORH) vs sonic action heads (SAH) toothbrushes. So I suggest that the title should be changed accordingly to be related to the aim of the article.
- Results:
Although the literature search is described in detail, the results in Table 1 should be included as supplemental material. The reason for exclusion could be summarized and included in figure 1 if the format of the study flow used was the Prisma 2009 format (Moher D, Liberati A, Tetzlaff J, Altman DG, The PRISMA Group (2009).
Table 6: Results from Zimmer are not reported.
Overall, the manuscript review and analysis of the articles conducted was very well done. I would suggest the authors register it as a systematic review at PROSPERO and, once approved, report the results following its guidelines as a systematic review.
Author Response
Dear reviewer, thank you for your precious suggestions. It was useful to improve the manuscript.
Introduction
We performed the modifications you requested in the aim statement leaving the title as it is.
Although the literature search is described in detail, the results in Table 1 are not included as supplemental material instead they are in the text because is an editorial request. The reason for exclusion are summarized and included in figure 1 according the guideline for narrative review.
Pag. 11, table 6: The results of Zimmer were not reported because they did not analyze the gingival index, instead the data are present in table 7 and 8 for Plaque Index and Bleeding on Probing.
Moreover, the article seems, to all intents and purposes, a systematic review, with PICO question, following PRISMA guidelines, but we called it "narrative" in the title and text because there are some bias, it is not registered on PROSPERO and the needs of authors are changed to a narrative review.

Reviewer 3 Report
Dear authors,
The article is interesting and deserves publication.
Just a few questions:
-The article is, to all intents and purposes, a systematic review, with PICO question, independent screening, risk of bias assessment and statistics. Why do you call it "narrative" in the title? The title needs to be changed.
-Like all systematic reviews, this too must be registered in PROSPERO, please submit the registration application and provide, in the text, the number, even if provisional.
-Why did you exclude studies comparing 2 or more different types of 18 sonic/roto-oscillating toothbrushes?
-Please provide inter and intra reviewer Agreement Q coefficient.
Thanks
Best regards
Author Response
Reviewer 3
I would like to thank the reviewer to have appreciated the article.
Yes, the article is, to all intents and purposes, a systematic review, with PICO question, independent screening, risk of bias assessment and statistics. I called it "narrative" in the title and text because there are some bias and the needs of authors are changed.
Pag. 3, line 96: We excluded studies comparing two or more different types of sonic toothbrushes and studies comparing two or more different types of oscillating/rotating toothbrushes because the aim of the present review is to compare oscillating/rotating heads (ORHs) vs sonic action heads (SAHs), it is not our intent to study which ORHs is better compared to one other ORHs or with SAHs is better compared to one other SAHs.
We calculated the intra reviewer inter Agreement Q coefficient as 90% and intra reviewer as 100%, but it is not a systematic review with metanalysis, instead it is not requested for a narrative review. Do you suggest adding it in the text?

Round 2
Reviewer 2 Report
The authors addressed my comments and made the suggested edits.
Reviewer 3 Report
Dear authors,
thank you for the answer. I'm not sure to understanding the presenceof "some bias and the needs of authors changed" as causal relation to the change from systematic to narrative, but I'll go with it. That's ok.
Regarding the two or more of the same type toothbrushes I guess you meant ONLY two or more of the same type, without comparison with a different type, which is ok as a reason for exclusion. However the sentence "Studies comparing 2 or more different types of 18 sonic/roto-oscillating toothbrushes were excluded" it's miselading. Juts make it cleared as when you explain it to me in the answer.
If you decided it doesnhave to be a systematic review, then no Q needed.
I don't need to revise the manuscript again, just change that little sentence before publication.
Congratulations.
This manuscript is a resubmission of an earlier submission. The following is a list of the peer review reports and author responses from that submission.